# Assessment of Student Pharmacists’ Co-Curricular Professionalization Using an Impact Scale

**DOI:** 10.3390/pharmacy12040117

**Published:** 2024-07-25

**Authors:** Laurie L. Briceland, Megan Veselov, Kelly Bach

**Affiliations:** Department of Pharmacy Practice, Albany College of Pharmacy and Health Sciences, Albany, NY 12208, USA

**Keywords:** co-curricular assessment, co-curricular impact scale, co-curriculum, pharmacy education, professional identity formation, student professional development, student professionalization

## Abstract

Co-curricular participation is a required component of the pharmacy program. Assessment of co-curricular activities has proven challenging due to lack of manpower to address the workload of reviewing multiple critical reflections. This project documented the professionalization impact of co-curricular involvement and secondarily explored the utility of our assessment tool, the Co-curricular Impact Scale (CIS), developed to streamline the assessment process. First- through third-professional-year students (P1, P2, P3) participated in five co-curricular domains: (i) professional development/education; (ii) patient care service; (iii) legislative advocacy; (iv) leadership/service to the pharmacy profession; and (v) healthcare-related community service. For the CIS, 16 questions were developed and mapped to 11 educational outcomes and included assessing the impact of immersing in an authentic learning experience, collaborating with healthcare professionals, and preparing for the pharmacist role. A group of 296 students rated the impact of participation as low, moderate, or significant for five events annually. Based on 717 entries, the two attributes deemed most impactful were: “Activity immersed me in an authentic learning experience” (95% ≥ Moderate Impact) and “Activity improved my self-confidence” (93% ≥ Moderate Impact). P1 students found slightly less impact in co-curricular participation (83.5%) than P2 (88.4%) and P3 (86.8%) counterparts. The CIS proved to be an efficient method to collate impact of co-curricular involvement upon student professionalization.

## 1. Introduction

A co-curricular requirement was introduced for Doctor of Pharmacy programs within the 2016 Accreditation Council for Pharmacy Education (ACPE) standards [1]. Broadly defined, the co-curriculum encompasses experiences that complement, enhance, and advance the required didactic and experiential curricula within the degree program, and promotes the growth of well-rounded learners [2]. Co-curricular experiences should be intentionally designed and aligned to the curriculum, particularly within the affective elements of the 2013 Center for Advancement of Pharmacy Education (CAPE) educational outcomes in Domain 3 (i.e., education, patient advocacy, interprofessional collaboration, cultural sensitivity, and communication), and Domain 4 (i.e., self-awareness, leadership, innovation and entrepreneurship, and professionalism) [3]. To comply with ACPE’s accreditation stipulation, all student pharmacists must engage in the co-curriculum, though students need not complete identical experiences but rather may personalize their co-curricular involvement to their own professional and personal goals [1]. Many colleges of pharmacy (including our own) have categorized co-curricular offerings into groupings from which students participate in a designated number of activities of their choosing over the didactic years of the curriculum; typical co-curriculum categories include professional education, patient care service, legislative advocacy, professional service/leadership, and healthcare-related community service [4,5,6,7].

Since the inception of the co-curriculum requirement, several studies [8,9,10] have documented strong connections between co-curricular engagement and professional identity formation (PIF). PIF is that evolutionary, longitudinal process in which student pharmacists are immersed into authentic practice scenarios (which are a hallmark of the co-curriculum); over time, students internalize the profession’s core values and beliefs and eventually begin to “think, act, and feel like a pharmacist” [11]. Arnoldi et al. reported that upon analysis of student self-reflections on professionalization impact within their pharmacy program, the co-curriculum emerged as highly valued and was cited most frequently by students as the most impactful program component in their professionalization [12]. Additionally, the co-curriculum can be instrumental in promoting academic success; one report indicated that students who are highly engaged in the co-curriculum are more likely to hold leadership roles, and those student leaders were more likely to earn higher grade point averages and reduced risk of academic deficiency when compared to less involved counterparts [13]. Another systematic review of co-curricular programs reported that in multiple studies, students perceived an increase in self-confidence and abilities through co-curricular participation [14].

Given all these positives upon adoption of a co-curriculum, are there any identified shortcomings? Maerten-Rivera et al. addressed this question by surveying accredited pharmacy programs to determine challenges in implementing the co-curriculum. The challenges most frequently reported by the 107 (75%) respondents were (i) closing the loop in terms of faculty providing feedback to students on their co-curricular participation (50.5%); (ii) assessing the co-curricular activities and plan (50.5%); and (iii) documenting how the student’s co-curricular activities advanced the learning (49.5%) [15]. These authors elucidated that the challenges were due to struggles with “co-curriculum burden”, linked to a lack of buy-in from faculty, staff, and students; insufficient staff/faculty time for the given cohort size; and/or a lack of clear definition of what constitutes “co-curriculum”. In a companion study by Maerten-Rivera et al., the most common assessment methodology to determine the impact of the co-curriculum was student self-reflection (89.7%), with a smaller proportion of programs employing self-assessment surveys (63.6%) and checklists (37.3%) [6]. Student self-reflection on the experiences is indeed a key component in optimizing co-curricular learning, as derived from the constructs of the continuing professional development cycle [16,17]. Yet, while self-reflection is the most employed assessment methodology, 50.5% of survey programs noted assessment as a challenge [15]. For instance, Hoffman et al. reported that while self-reflection was their primary method of assessment, manpower issues precluded them from being able to review each individual reflection, and instead they performed aggregate assessments; they suggested the “need to re-evaluate the reflection process and create a more structured analysis” of the co-curricular outcomes [4]. Vos et al. reported that their lengthy annual reflection methodology provided little benefit to both students who wrote them and faculty who reviewed them, and thus they replaced the methodology with co-curricular goal setting [17]. Mekonnen et al. [8] reported a lack of ability to achieve a 100% completion rate for student reflections and attributed this to a lack of dedicated personnel for oversight of the co-curriculum, and difficulties in developing and enforcing consequences for lack of submission. We too experienced similar struggles in providing feedback and documenting and assessing our own co-curriculum.

We aspired to create an effective, more streamlined approach to assess students’ co-curricular activity, such as an assessment scale tool; this methodology might address the noted shortcomings of self-reflections by simplifying the documentation (for students) and assessment of learning process for both students and faculty. Also, a simpler approach might lead to greater completion rates and increase the likelihood that the faculty reviewers would provide timely feedback, addressing the “closing the loop” shortcoming. A review of the literature revealed a study by Matthews et al. [18], who reported the use of an end-of-semester assessment survey tool to evaluate co-curricular participation; this tool was used to determine whether students experienced improvement in the program’s six overarching ability-based outcomes, two of which were professionalism and engaging in personal and professional development. Their survey tool did document students’ perceptions of improvement in these two broad domains; we aimed to build upon this type of co-curricular assessment tool by creating a more focused impact scale mapped primarily to the affective domains (3 and 4) of the CAPE outcomes. The primary aim of this study was to document the impact of co-curricular participation upon student pharmacists’ professionalization; the secondary aim was to explore the utility of our Co-curricular Impact Scale as a potential streamlined approach to assessing co-curricular professionalization. 

## 2. Materials and Methods

This project was conducted at Albany College of Pharmacy and Health Sciences (ACPHS), a private college in New York State (USA) with typical class sizes of 80 to 130 students in the 4-year Doctor of Pharmacy program. The project underwent Institutional Review Board review and met the criteria for exemption from requirements of federal regulations.

### 2.1. Description of Our Co-Curricular Plan

We established our Pharmacist-in-Training (PhIT) Portfolio in 2015 to serve as our Co-curricular Plan, housed in our Canvas learning management system (LMS). The PhIT Co-curriculum Subcommittee (“Subcommittee”) of the Pharmacy Experiential Education Committee oversees the co-curriculum. The details of the co-curricular requirements and oversight are shown in Table 1.

Prior to the 2021–2022 academic year, our co-curricular assessment modality was student self-reflection of each co-curricular event uploaded in the PhIT Portfolio. The expectation was that Subcommittee members (or subsequently faculty advisors, as the cohort size proved too large for the Subcommittee to handle all reviews) were to review and provide feedback to the students on their reflections. However, this assessment methodology proved time-consuming for students, which may be a contributor to PhIT Portfolio completion rates of ~75%; additionally, the faculty review process (by advisor or Subcommittee member) was not only labor-intensive but occurred sporadically, with feedback provided months after the event participation if at all. Thus, the Subcommittee determined that a process improvement to assess co-curricular participation was necessary, which led to the development of the Co-curricular Impact Scale.

### 2.2. Description of Co-Curricular Impact Scale (CIS)

The Subcommittee created and implemented a Co-curricular Impact Scale (CIS), housed within our LMS, as a tool for students to more efficiently document and assess the impact of participation in the co-curriculum. A review of the literature led us to the work of Gettig et al. [19], in which a co-curricular content analysis was performed and mapped to 12 CAPE outcomes [3]; this study determined the potential for each co-curricular offering to provide students up to 12 professionalization opportunities. Building on this work, we created our CIS with the overarching intent to employ the assessment tool to document the professionalization that was predicted to occur per the Gettig et al. paper [19]. Specifically, we adapted and expanded their list of professionalization attributes to include 16 items which students would assess for co-curricular impact; each of these attributes was mapped to at least one of 11 CAPE outcomes, primarily in Domains 3 and 4 [3], as shown in Table 2.

Upon participating in a co-curricular event, students attested to their attendance and documented the activity under the applicable co-curricular category in the PhIT Portfolio by typing in the name and date of the event. Using the CIS, students assessed each of the 16 attributes (Table 1) in terms of impact on professionalization, utilizing a 3-point Likert scale format, ranging from low to moderate to significant impact, or not applicable. P1–P3 students completed a CIS assessment for each of their 5 self-selected co-curricular events annually. The CIS assessment replaced guided self-reflection in P1 and P2; one final P3 reflection remained. The Faculty Advisor received an individualized progress report for each P2 advisee from the Subcommittee to discuss progress with PhIT Portfolio requirements. The Subcommittee compiled a year-end aggregate report on CIS results and provided individual feedback on the P3 rubric-assessed reflections. The Co-curricular Professional Development Plan remained a component of the PhIT Portfolio. 

### 2.3. Assessment of Impact of the CIS on Students’ Professionalization

For this evaluation, the 2023–2024 academic year PhIT Portfolio upload data were assessed. “Completion” of the PhIT requirements for a given student was defined as uploading all 5 CIS assessment uploads into the PhIT Portfolio. One investigator (MV) de-identified CIS data and downloaded them into an Excel database for the authors’ use to determine the impact of the co-curriculum on student professionalization. Specifically, descriptive statistics were used to describe the professionalization impact of co-curricular participation based on class year (P1–P3), the 5 co-curricular event categories, and the 16 professionalization attributes that were assessed with the CIS. For purposes of this analysis, the Likert scale ratings of moderate and significant impact were combined; the rationale for the combination of ratings was to demonstrate that students gained at least some noticeable benefit (i.e., “moderate”) as opposed to the “low impact” in which students would have received minimal benefit.

## 3. Results

During the 2023–2024 cycle, a total of 717 co-curricular events were entered into the PhIT Portfolios by 296 students. This translates to a PhIT Portfolio completion rate for the five required uploads/CIS assessments of 74% of 75 P1 students, 87% of 90 P2 students, and 81% of 131 P3 students. Question #16 on the CIS was inadvertently omitted in the online PhIT Portfolio, and thus data exist only for Questions 1–15 of the CIS.

### 3.1. Class Year Comparison of Impact of Co-Curricular Involvement

P1, P2, and P3 students participated in events in their three required (and two students’-choice) co-curricular categories. The percentage of students who assessed the professionalization impact of their participation in required events to be of at least moderate impact is shown in Table 3.

As per Table 3, a slightly smaller percentage of P1 students (83.5%) found co-curricular participation in their required categories to be impactful as compared to P2 (88.4%) and P3 (86.8%) students, attributed to lower ratings on the healthcare-related community service event for P1 students.

### 3.2. Comparison of Professionalization Impact among Five Co-Curricular Domains of the PhIT Portfolio

Figure 1 shows the distribution of professionalization impact for all P1–P3 students (N = 296) based on CIS assessment; this includes all five PhIT uploads per student (i.e., required domains plus students’-choice domains), spanning each of the five co-curricular domains.

While all categories showed at least moderate impact in >86% of students, the two most impactful categories were leadership/service to the profession (96.6%), and professional development (95.6%). The least impactful category was healthcare-related community service (86.8%). 

### 3.3. Comparison of Impact among Professionalization Attributes Assessed in the Co-Curricular Impact Scale (CIS) 

Data were available for analysis for Questions 1–15 of the CIS. Based on Likert scale ratings, items rated as having moderate or significant professionalization impact were grouped together and reported in Table 4. The top four ratings in each category are highlighted in yellow.

As per Table 4, our CIS compilation revealed that across all five co-curricular domains, “immersion into authentic learning experiences”, and “increase in self-confidence” were top-four professionalization impacts for the 296 students in the cohort. For question 6: *This activity was of high quality and will be referenced in my future professional interactions*, the highest impact was in leadership/service to the profession (95.9%), and the lowest impact in healthcare-related community service (78.3%). The CIS provided a much greater amount of data than could be aggregated under specific CAPE outcomes, or specific class year (data not shown). 

## 4. Discussion

### 4.1. Documenting and Assessing Co-Curricular Professionalization Using Co-Curricular Impact Scale

The inclusion of co-curricular participation within the pharmacy degree program is crucial to the advancement of PIF [10,11,20], especially given that pharmacy students may not perceive didactic experiences to be impactful in supporting PIF, attributed to the lack of authenticity within the exercises [13,20]. The co-curriculum is renowned for providing highly valued authentic practice experiences [13], which was corroborated in our evaluation using the CIS (Table 4, Question 1); across all five co-curricular categories, students gave top-four impact ratings for the authenticity of the co-curricular activity. The CIS was also able to capture another expectation of the co-curriculum in the self-awareness CAPE domain by increasing students’ self-confidence [14]; this is demonstrated in Table 4, Question 2, with students again reporting top-four impact ratings across all five co-curricular categories. 

It is important to have an effective assessment method for students to document co-curricular impact and for faculty to review and provide prompt feedback. As mentioned in the Introduction, Maerten-Rivera et al. determined that student self-reflection is the most common assessment methodology in practice (89.6% survey respondents) [6]; the narrative approach inherent in critical reflection does remain the hallmark to “support students to derive personal and professional meaning, learn from their experiences, and make PIF explicit” [20]. However, others have described the shortcomings of exclusively using reflections as the assessment methodology [4,17], such as repetition of the reflection process leading to little benefit to students, and the lack of manpower to review the essays; these authors suggest alternative approaches to assessment. We agreed that our original requirement that each student was to complete five reflections per year was too cumbersome for students. We based this contention on the overall curricular and co-curricular “reflection burden”, in which our students were completing at least 50 reflections per year [21], and on PhIT Portfolio completion rates no higher than 75% per cohort. Likewise, we found the requirement too burdensome for faculty, as the faculty reviews were sporadically completed.

Our CIS served to streamline the assessment process for P1–P3 students by removing 14 of 15 reflections (as we kept one final reflection in P3), and inserting the 16-item CIS, to be completed for 15 events. The CIS assessments very aptly served to provide evidence of the professionalization impact of co-curricular participation, as shown in Table 4. While we did not directly complete a head-to-head comparison of self-reflections versus CIS in terms of which methodology proved more efficient, we believe the CIS to be a more efficient assessment tool from the faculty perspective. We base this contention on the reduction in workload for faculty (whether it was the Faculty Advisor or a Subcommittee faculty member) by removing the 14 reflections per student and need for reviewing/providing feedback on each. Also, the CIS facilitated ease of reporting aggregate results of co-curricular impact on student professionalization. With the introduction of the CIS, we did not entirely abandon critical self-reflection in our Co-curricular Plan, as we retained one critical reflection in P3, and retained the Co-curricular Professional Development Plans (P1 and P2); faculty provided feedback for each of those assignments. Thus, in our experience, keeping a few self-reflection components, in conjunction with the CIS, provides a more streamlined, yet comprehensive, assessment approach to the co-curriculum.

While we did not directly survey students in terms of their perspectives on the CIS, we were very satisfied with the P2 and P3 PhIT Portfolio completion rates (87% and 81%, respectively), which were higher than our historical ~75% completion rate using self-reflections. The P1 completion rate remained essentially unchanged (74%) compared to our historical ~75% completion rate. Anecdotally, through faculty discussions with P1 students, and through reading P1 Co-curricular Development Plans, some P1 students purposefully chose not to allocate time on the co-curriculum, as they aimed to focus on their didactic academic coursework as they settled into the pharmacy program. This lower completion rate of the P1 class provides an opportunity for the Subcommittee to work with P1 faculty advisors, students, and academic advising/counseling services, to emphasize the importance of the co-curriculum in professional development, beginning in P1, yet provide the necessary resources to P1 students to promote student success. Some additional methods to encourage completion for any class year include providing class time for students to complete their CIS (provided they had attended the events), or perhaps holding students accountable if they do not complete the required PhIT Portfolio, which has proved for us to be a difficult undertaking in developing and enforcing consequences, as corroborated by others [8].

### 4.2. Limitations and Future Directions

There are limitations to our work. As mentioned in Section 4.1, students were not surveyed about their perspectives on the CIS and thus we cannot verify the readability or understandability of the CIS statements and Likert ratings, such as the distinction between low, moderate, and significant impact. Likewise, we cannot definitively state that the CIS is a preferred methodology from the students’ viewpoint. To address these shortcomings, in the future, we could consider hosting a focus group or surveying students concerning perspectives on the CIS. Additionally, we can add instructions as to the definitions of the Likert ratings (low, moderate, and significant impact). It would also be useful to consider validating the survey instrument, which could garner faculty input and improve the generalizability of the CIS tool. Also, CIS entries within the PhIT Portfolio are based on student self-report, a process commonly used by others [5,17], with no verification of student attendance at events. Thus, it is possible that a subset of students is falsifying co-curricular involvement uploads, even though students attest to attendance. Efforts to address verification of attendance have been periodically attempted, including using a student swipe-in mechanism or student sign-in to prove attendance at a co-curricular event, but nothing has been procedurally formalized. The barrier that we have faced in enacting an attendance verification mechanism has been the lack of manpower to administer and track this information, given the dozens of events occurring annually; this manpower issue is a common barrier across colleges of pharmacy [15]. Lastly, the quality of each co-curricular activity is not formally assessed by the Subcommittee, or any other body; thus, it is possible that the reason why the quality of some categories is rated lower than others is that, in fact, the programs offered may be subpar. For future directions, we plan to drill down into the volumes of data documented within the CIS and make activity-specific suggestions. For instance, we can identify specific activities that received the low ratings (e.g., <65% impact) on quality (Question 6 on the CIS) and suggest to OSEW/POC to revise or omit the activities; at the same time, we can offer a report to these co-curriculum sponsors on those activities that are deemed of high quality, re-enforcing their merits. 

## 5. Conclusions

Our cohort of 296 students participated in five co-curricular categories including professional development, legislative advocacy, healthcare-related community service, patient care, and leadership. We have introduced a Co-curricular Impact Scale (CIS) assessment tool to document and assess the professionalization impact of co-curricular involvement. Using the CIS tool, students reported that for all five co-curricular categories, “immersion into authentic practice settings” and “increasing self-confidence” were both in the top four areas of impact. P1 students found slightly less impact in co-curricular participation than P2 and P3 counterparts. Faculty overseeing the co-curriculum deemed the CIS to be a more efficient co-curricular assessment tool (than using solely critical reflections), in terms of reducing faculty and student workload, while still effectively measuring the professionalization impact of the co-curriculum. In our experience, this CIS assessment tool proved to be an efficient method of co-curricular assessment. 

## Figures and Tables

**Figure 1 pharmacy-12-00117-f001:**
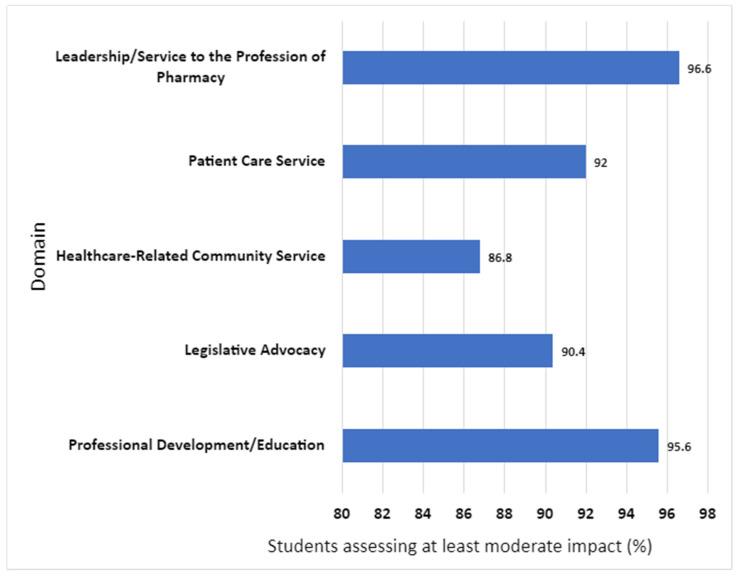
Distribution of Professionalization Impact For Co-curricular Domains (P1–P3 Students combined).

**Table 1 pharmacy-12-00117-t001:** Description of the Co-curricular Plan.

Pharmacist-in-Training (PhIT) Portfolios
Oversight: PhIT Co-curriculum Subcommittee of the Pharmacy Experiential Education Committee (“Subcommittee”) oversees the co-curriculum and is charged to: Annually review and update as needed the co-curricular requirements for P1–P3 students Communicate the PhIT Portfolio Co-curricular Plan to students and faculty advisors Establish and maintain a mechanism for students to upload documents and assessment of impact of co-curricular involvement Liaise with co-curricular sponsors to determine which activities are “PhIT-approved” Assess/provide feedback to students on the educational impact of the students’ participation in the co-curriculum
Requirements: each student is expected to document participation in 5 self-selected co-curricular activities annually; of these, 3 domains are designated as required and 2 are the student’s choice, which can be selected from any of the 5 domains (differs slightly for P1–P3 students: see Appendix A: Pharmacist-in-Training (PhIT) Portfolio Requirements 2023–2024): Professional development Patient care service Legislative advocacy Leadership/service to the pharmacy profession Healthcare-related community service
Co-curricular Programming: Creative programming is offered collaboratively through Student Affairs’ Office of Student Engagement and Wellness (OSEW) and the College’s Professional Organization Council (POC). Student leaders introduce organizations and co-curricular programming to students during orientation and bi-annual involvement fairs. Through the Engage platform, weekly announcements provide students with information and registration links for upcoming co-curricular events depicted by the College’s PhIT-approved stamp.
Curricular Component: In 2020, a Co-curricular Professional Development Plan (CPDP) assignment was instituted within the P1 Foundations of Pharmacy course to assist students with tailoring co-curricular participation to their own specific goals. Employs self-reflection in having students identify their overarching co-curricular goals and creation of specific plans with timeline to execute. Students receive feedback on the CPDP from P1 course faculty. In year P2 the CPDP is revisited in the Patient Care Workshop course.

**Table 2 pharmacy-12-00117-t002:** Professionalization domains included in Co-curricular Impact Scale assessment tool.

Professionalization Attribute Assessed	CAPE Mapping
This activity immersed me in an authentic learning experience.	2.1, 2.3, 3.2
This activity improved my self-confidence in providing patient care, advocacy, community service, and/or leadership/service to the profession.	4.1
This activity improved my understanding and/or abilities to collaborate with other healthcare professions.	3.4
This activity allowed me to interact with patients and practice culturally sensitive care. This activity opened my eyes to provide care to patients who have cultural competency within pharmacy practice.	3.5
This activity allowed me to improve my communication skills with patients, colleagues, and/or other healthcare providers.	3.6
This activity was of high quality and will be referenced in my future professional interactions.	4.1
This activity made me feel more prepared for APPEs and becoming a pharmacist.	4.1
This activity increased my self-awareness such that I can more easily identify my strengths and weaknesses.	4.1
This activity coaxed me out of my “comfort zone” and promoted personal/professional growth.	4.1
This activity has opened my eyes to ideas/perspectives not previously recognized and stimulated intellectual curiosity.	4.1
This activity exposed me to potential career opportunities.	4.1
This activity will help me gain competency and facilitate life-long learning.	4.1
This activity allowed me to develop and refine my leadership abilities and skills.	4.2
This activity broadened my professional horizons and perspectives in the areas of professionalism, altruism, accountability, and/or integrity.	4.4
This activity improved my academic/clinical knowledge related to pharmacy practice.	1.1
This activity allowed me to apply creativity, entrepreneurship, and/or an innovative mindset to address challenges and promote positive change.	4.3

CAPE = Center for Advancement of Pharmaceutical Education [3]. 1.1 = Learner; 2.1 = Patient-centered care; 2.3 = Health and wellness; 3.2 = Educator; 3.4 = Collaborator; 3.5 = Cultural Sensitivity; 3.6 = Communicator; 4.1 = Self-aware; 4.2 = Leader; 4.3 = Innovator; 4.4 = Professional.

**Table 3 pharmacy-12-00117-t003:** Students’ evaluation of professionalization impact of co-curricular participation as assessed using Co-curricular Impact Scale.

Co-curricular Uploads of Required Domains per PhIT Portfolio Requirements	P1 (Class of 2027) N = 75 Moderate and Significant Impact Ratings (% of Students)	P2 (Class of 2026) N = 90 Moderate and Significant Impact Ratings (% of Students)	P3 (Class of 2025) N = 131 Moderate and Significant Impact Ratings (% of Students)
**Professional Development/** **Educational Event**	88.2	91.9	89.3
**Legislative Advocacy Event**	85.4	81.1	84.6
**Healthcare-related Community Service Event**	77.2	X	X
**Leadership/Service to the Profession Event **	X	92.7	X
**Patient Care Service**	X	X	86.9
**Summary Impact**	83.5	88.4	86.8

PhIT = Pharmacist-in-Training Portfolio. P1 = first professional year. P2 = second professional year. P3 = third professional year. X = students’-choice uploads; only required uploads were included in Table 2 data comparisons.

**Table 4 pharmacy-12-00117-t004:** Professionalization impact of co-curricular participation as assessed by Co-curricular Impact Scale for combined P1, P2, and P3 students (N = 296).

Attribute Assessed	Moderate and Significant Impact Ratings (% of Students)				
	Co-curricular Category (N = # of Events)				
	Professional Development N = 247	Legislative Advocacy N = 240	Healthcare-Related Community Service N = 55	Patient Care Service N = 98	Leadership/Service to Profession N = 77
1. This activity immersed me in an authentic learning experience.	98.7	97.1	86.7	92.3	98.6
2. This activity improved my self-confidence in providing patient care, advocacy, community service, and/or leadership/service to the profession.	95.9	89.7	89.6	93.9	96.1
3. This activity improved my understanding and/or abilities to collaborate with other healthcare professions.	93.4	82.8	80.9	82.9	89.9
4. This activity allowed me to interact with patients and practice culturally sensitive care. This activity opened my eyes to provide care to patients who have cultural competency within pharmacy practice.	75.3	77.5	78.1	88.4	92.7
5. This activity allowed me to improve my communication skills with patients, colleagues, and/or other healthcare providers.	85.9	75.3	85.7	92.6	94.3
6. This activity was of high quality and will be referenced in my future professional interactions.	92.4	85.1	78.3	88.3	95.9
7. This activity made me feel more prepared for APPEs and becoming a pharmacist.	92.7	80.0	73.8	84.4	93.0
8. This activity increased my self-awareness such that I can more easily identify my strengths and weaknesses.	86.2	79.6	63.3	85.6	95.7
9. This activity coaxed me out of my “comfort zone” and promoted personal/professional growth.	86.0	86.5	76.0	88.3	94.7
10. This activity has opened my eyes to ideas/perspectives not previously recognized and stimulated intellectual curiosity.	94.4	75.8	85.1	87.2	86.5
11. This activity exposed me to potential career opportunities.	91.8	88.5	55.8	73.0	84.6
12. This activity will help me gain competency and facilitate life-long learning.	86.2	75.7	69.6	85.6	94.4
13. This activity allowed me to develop and refine my leadership abilities and skills.	80.7	84.6	82.2	84.9	94.8
14. This activity broadened my professional horizons and perspectives in areas of professionalism, altruism, accountability, and/or integrity.	84.5	83.3	84.1	88.9	93.3
15. This activity improved my academic/clinical knowledge related to pharmacy practice.	90.0	83.6	68.3	82.4	84.9

Yellow highlight = Top-four professionalization impact.

## Data Availability

Data are contained within the article.

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
