# Peer review of "Assessment of Student Pharmacists’ Co-Curricular Professionalization Using an Impact Scale"

_pharmacy, 2024, doi:10.3390/pharmacy12040117_

Round 1
Reviewer 1 Report
Comments and Suggestions for Authors
Thank you for the opportunity to review this manuscript. I would like to commend the authors for a thorough review of the literature within the manuscript as well as identifying an opportunity to solve a problem that is contained both within the literature and their own program. The writing is a strength of the manuscript.
Please consider some following comments as you revise your manuscript:
1) Given that this is a scale (and submitted as a research manuscript), what type of approach did you take to pre-test or obtain validity evidence? These are important considerations for the utility and accuracy of your instrument as well as the generalizability. If this was not done, please include this as a limitation.
2) Consider rethinking your study objective. It describes utility, but the data you collected is primarily the outcomes of the survey. Should utility be a secondary objective? There was only one paragraph in the results on the utility, most of which was anecdotal. This is also not described in the methods either. Please take a look at the whole manuscript and think through balancing the data you obtained and how you have organized your objective, methods, and results.
3) Consider condensing your co-curricular plan overview in the methods, as it is not the driver of the study. The development and implementation of the assessment tool is the driver. One approach to condensing it would be to consider outlining it briefly in a table.
4) In 2.3, you mention data analysis and comparisons. It looks like you ran descriptive statistics only. Please make sure you are clear with the analytic approach in this section. Further, you could consider some inferential statistics to truly determine if there are differences.
5) Now that COEPA and the new ACPE Standards have been finalized, how will that impact your future directions and conclusions?
Author Response
1. Summary |
|
|
||||||
Thank you very much for taking the time to review this manuscript. Please find the detailed responses below and the corresponding revisions/corrections highlighted/in track changes in the re-submitted files.
|
||||||||
2. Questions for General Evaluation |
Reviewer’s Evaluation |
Response and Revisions |
||||||
Does the introduction provide sufficient background and include all relevant references? |
Yes/Can be improved/Must be improved/Not applicable |
|
||||||
Are all the cited references relevant to the research? |
Yes/Can be improved/Must be improved/Not applicable |
|
||||||
Is the research design appropriate? |
Yes/Can be improved/Must be improved/Not applicable |
Improvements noted in revision |
||||||
Are the methods adequately described? |
Yes/Can be improved/Must be improved/Not applicable |
Improvements noted in revision |
||||||
Are the results clearly presented? |
Yes/Can be improved/Must be improved/Not applicable |
|
||||||
Are the conclusions supported by the results? |
Yes/Can be improved/Must be improved/Not applicable
|
|
||||||
3. Point-by-point response to Comments and Suggestions for Authors Thank you for the opportunity to review this manuscript. I would like to commend the authors for a thorough review of the literature within the manuscript as well as identifying an opportunity to solve a problem that is contained both within the literature and their own program. The writing is a strength of the manuscript. Thank you for these positive comments.
Please consider some following comments as you revise your manuscript: |
||||||||
Comments 1: Given that this is a scale (and submitted as a research manuscript), what type of approach did you take to pre-test or obtain validity evidence? These are important considerations for the utility and accuracy of your instrument as well as the generalizability. If this was not done, please include this as a limitation. |
||||||||
Response 1: Thank you for pointing this out. We did not pre-test or validate the survey instrument and agree with you that this is a study limitation in terms of readability and generalizability of using the CIS tool. Therefore, we have added a statement regarding this issue in 4.2 (Limitations and Future Directions), paragraph 1 in the manuscript revision. We also tempered the final sentence of the conclusions to minimize the generalizability of the unvalidated CIS.
|
||||||||
Comments 2: Consider rethinking your study objective. It describes utility, but the data you collected is primarily the outcomes of the survey. Should utility be a secondary objective? There was only one paragraph in the results on the utility, most of which was anecdotal. This is also not described in the methods either. Please take a look at the whole manuscript and think through balancing the data you obtained and how you have organized your objective, methods, and results. |
||||||||
Response 2: Thank you for this insightful comment. Upon rethinking the project object, we agree with your point that the professionalization outcomes are the primary project objective, with the utility of the CIS being of secondary importance. Accordingly, we have modified the manuscript by revising the title, abstract/methods, introduction/aims, and conclusion.
|
||||||||
4. Response to Comments on the Quality of English Language |
||||||||
|
||||||||
(Not applicable)
|
||||||||
5. Additional clarifications |
||||||||
None |

Reviewer 2 Report
Comments and Suggestions for Authors
The manuscript aims to presents newly developed Co-curricular Impact scale (CIS) and how CIS scale can be used to simplify student assessment following co-curricular activities. The study develops further the work of Gettig and coworkers from 2020 concerning mapping various co-curricular activities to ACPE standards. The authors present the results of self-assessment of a group of 296 students representing 3 years of Pharmacy programme, who attended a total of 717 co-curricular activities. The presented results show that the proposed method of assessment is useful in assessing a variety of activities representing different CAPE categories and can be more efficient as compared to reflection. The main strength of the manuscript are as follows: a clear structure of the article, well described research gap and a choice of topic that is significant to educators in the field of Pharmacy. Data presentation is clear and consistent, the article is well written. The conclusions are consistent with the presented evidence, cited literature is up to date.
I would raise some points that could be better addressed by authors:
1) In tables 1 and 3, there is a description of subdomain 4.3 – Innovator. However CIS tool does not contain any question regarding this subdomain. Why was this area not addressed?
2) The CIS tool is under development, e.g. authors stated that students perspective concerning the tool was not presented. Moreover, opinion of faculty staff could also be investigated regarding the efficacy of this new tool. Authors could describe what are implications for further research and how this gap could be filled.
3) The most relevant previous studies concerning the co-curricular activities are: the work of Matthews from 2022 – reference 18 ( https://doi.org/10.1016/j.cptl.2022.02.003 ) and the study of Pahl from 2022 – reference 10 (https://doi.org/10.1016/j.cptl.2022.04.016 ). They investigated e.g. the impact of co-curricular activities on learning outcomes and its progression throughout the study years as well as the value of co-curricular activities in professional identity enhancement and professional advancement of the students. In the Section 4.1. authors could discuss their results with Matthews and Pahl findings.
4) The authors highlighted the most impactful categories in Table 3. However they did not address the differences between them and the other, less impactful categories. This should be picked up in the Section 4.1.
5) Although, the data is compilation of students’ experiences out of numerous different smaller activities, there is some observable difference between 5 categories, as presented in Fig. 1. It could be interesting for the readers if authors tried to explain why ‘Health-related community services’ yielded the least impact out of all examined categories.
I would also point out some minor issues:
1) In table 3 column head ‘CAPE Mapping’ should be included to match table 1
Author Response
1. Summary |
|
|
||||||||||||||
Thank you very much for taking the time to review this manuscript. Please find the detailed responses below and the corresponding revisions/corrections highlighted/in track changes in the re-submitted files.
|
||||||||||||||||
2. Questions for General Evaluation |
Reviewer’s Evaluation |
Response and Revisions |
||||||||||||||
Does the introduction provide sufficient background and include all relevant references? |
Yes/Can be improved/Must be improved/Not applicable |
No issues to address in this section |
||||||||||||||
Are all the cited references relevant to the research? |
Yes/Can be improved/Must be improved/Not applicable |
|
||||||||||||||
Is the research design appropriate? |
Yes/Can be improved/Must be improved/Not applicable |
|
||||||||||||||
Are the methods adequately described? |
Yes/Can be improved/Must be improved/Not applicable |
|
||||||||||||||
Are the results clearly presented? |
Yes/Can be improved/Must be improved/Not applicable |
|
||||||||||||||
Are the conclusions supported by the results? |
Yes/Can be improved/Must be improved/Not applicable
|
|
||||||||||||||
3. Point-by-point response to Comments and Suggestions for Authors The manuscript aims to presents newly developed Co-curricular Impact scale (CIS) and how CIS scale can be used to simplify student assessment following co-curricular activities. The study develops further the work of Gettig and coworkers from 2020 concerning mapping various co-curricular activities to ACPE standards. The authors present the results of self-assessment of a group of 296 students representing 3 years of Pharmacy programme, who attended a total of 717 co-curricular activities. The presented results show that the proposed method of assessment is useful in assessing a variety of activities representing different CAPE categories and can be more efficient as compared to reflection. The main strength of the manuscript are as follows: a clear structure of the article, well described research gap and a choice of topic that is significant to educators in the field of Pharmacy. Data presentation is clear and consistent, the article is well written. The conclusions are consistent with the presented evidence, cited literature is up to date. Thank you for the positive commentary on our manuscript.
|
||||||||||||||||
Comments 1: In tables 1 and 3, there is a description of subdomain 4.3 – Innovator. However CIS tool does not contain any question regarding this subdomain. Why was this area not addressed? |
||||||||||||||||
Response 1: Thank you for pointing this out. We had in fact modified Question 16 in the CIS for the 2023-24 cycle to include a question mapped to CAPE 4.3 (See CIS document uploaded in Supplemental documents). In error, the old question 16 was included in the text of the manuscript. The correct question is now listed as Question 16 in Table 1 (now numbered Table 2). It is not included in Table 3 (now numbered Table 4), because question 16 was not activated in the LMS, so we do not list it in results. |
||||||||||||||||
Comments 2:  The CIS tool is under development, e.g. authors stated that students perspective concerning the tool was not presented. Moreover, opinion of faculty staff could also be investigated regarding the efficacy of this new tool. Authors could describe what are implications for further research and how this gap could be filled. |
||||||||||||||||
Response 2: Agree with these good points. In section 4.2 (Limitations), paragraph 1, we mention that in the future we could study the perspective of students. We have also added a statement on considering validating the instrument, to address faculty perspective.
|
||||||||||||||||
4. Response to Comments on the Quality of English Language |
||||||||||||||||
|
||||||||||||||||
Response 1: No issues
|
||||||||||||||||
5. Additional clarifications |
||||||||||||||||
None |

Reviewer 3 Report
Comments and Suggestions for Authors
The topic if the manuscript is important and interesting.
The abstract provides a good summary of the manuscript’s content.
Introduction section: is recommended a very short presentation of the Pharmacy Education System, to understand the knowledge on which it is based Co-curricular Professionalization
The material and method section details all information needed to understand the research process. Although, the used 4-point Likert scale, if the “not applicable” answer is included, is actually like a 3-point scale, so “moderate impact” is actually equivalent to an impact of about 33-66%, which makes questionable any firm conclusions. Since the research has been done, this use of the Likert skale in this way needs to be explained.
Row 47: instead of „including our own” – the name, county, city of the Collage must be named
The quality of Figure 1 should be improved
Because the Table 3 shows Moderate and Significant Impact Ratings (not only significant), is recommended to shows the scores in a chart format (e.g. Likert Scale Chart)
Information in the Table 1 are repeated in the Table 3. It is recommended numbering the statements cand the corresponding numbers to be included in the Table 3.
Discussion section: PIF abbreviation appears in row 54 also.
Limitations and Future Directions: it is recommended to include also obtain information on behalf of the employers or alumni regarding the impact of the Co-curricular activities.
References are relevant to the study and properly cited.
Author Response
1. Summary |
|
|
||||||||||||||||||
Thank you very much for taking the time to review this manuscript. Please find the detailed responses below and the corresponding revisions/corrections highlighted/in track changes in the re-submitted files.
|
||||||||||||||||||||
2. Questions for General Evaluation |
Reviewer’s Evaluation |
Response and Revisions |
||||||||||||||||||
Does the introduction provide sufficient background and include all relevant references? |
Yes/Can be improved/Must be improved/Not applicable |
|
||||||||||||||||||
Are all the cited references relevant to the research? |
Yes/Can be improved/Must be improved/Not applicable |
|
||||||||||||||||||
Is the research design appropriate? |
Yes/Can be improved/Must be improved/Not applicable |
Addressed in revised manuscript |
||||||||||||||||||
Are the methods adequately described? |
Yes/Can be improved/Must be improved/Not applicable |
Addressed in revised manuscript |
||||||||||||||||||
Are the results clearly presented? |
Yes/Can be improved/Must be improved/Not applicable |
|
||||||||||||||||||
Are the conclusions supported by the results? |
Yes/Can be improved/Must be improved/Not applicable
|
Addressed in revised manuscript |
||||||||||||||||||
3. Point-by-point response to Comments and Suggestions for Authors |
||||||||||||||||||||
The topic if the manuscript is important and interesting. The abstract provides a good summary of the manuscript’s content. Thank you for these positive affirmations.
Comments 1: Introduction section: is recommended a very short presentation of the Pharmacy Education System, to understand the knowledge on which it is based Co-curricular Professionalization |
||||||||||||||||||||
Response 1: The basis for the addition of a co-curricular requirement in the Doctor of Pharmacy program is included (with citations) in Section 1 (Introduction), paragraph 1; the benefits of the co-curriculum in terms of professionalization (Professional Identity Formation), with citations, are included in paragraph 2. No additional changes were made to the revision. |
||||||||||||||||||||
Comments 2: The material and method section details all information needed to understand the research process. Although, the used 4-point Likert scale, if the “not applicable” answer is included, is actually like a 3-point scale, so “moderate impact” is actually equivalent to an impact of about 33-66%, which makes questionable any firm conclusions. Since the research has been done, this use of the Likert skale in this way needs to be explained. |
||||||||||||||||||||
Response 2: Thank you for bringing this point to our attention; we have thus modified the scale from a 4-point to a 3-point Likert Scale, (with a “not applicable” option). This change is noted in the final paragraph of Methods 2.3. We elected to group moderate and significant ratings to demonstrate that students gained some noticeable benefit (i.e. “moderate”) as opposed to the “low impact” in which students would have received minimal benefit. We realize that this was not fully explained to students in our learning management system, which is a limitation of our work. We have added some verbiage to Section 4.2 (Limitations/Future Directions) to discuss this limitation and strategies to address this in the future. |
||||||||||||||||||||
4. Response to Comments on the Quality of English Language |
||||||||||||||||||||
Point 1: No issues |
||||||||||||||||||||
Response 1: No issues
|
||||||||||||||||||||
5. Additional clarifications |
||||||||||||||||||||
none |

Round 2
Reviewer 1 Report
Comments and Suggestions for Authors
Thank you for revising your manuscript. I have no further edits at this time (although, I believe you have a typo in the limitations - double-check sentences 5/6.